# The Interaction of Hydrogen with the van der Waals Crystal *γ*-InSe

**DOI:** 10.3390/molecules25112526

**Published:** 2020-05-28

**Authors:** James Felton, Elena Blundo, Sanliang Ling, Joseph Glover, Zakhar R. Kudrynskyi, Oleg Makarovsky, Zakhar D. Kovalyuk, Elena Besley, Gavin Walker, Antonio Polimeni, Amalia Patané

**Affiliations:** 1School of Physics and Astronomy, University of Nottingham, Nottingham NG7 2RD, UK; zakhar.kudrynskyi@nottingham.ac.uk (Z.R.K.); oleg.makarovsky@nottingham.ac.uk (O.M.); 2Advanced Materials Research Group, Faculty of Engineering, University of Nottingham, Nottingham NG7 2RD, UK; sanliang.ling@nottingham.ac.uk (S.L.); gavin.walker@nottingham.ac.uk (G.W.); 3Dipartimento di Fisica, Sapienza Università di Roma, 00185 Roma, Italy; elena.blundo@uniroma1.it (E.B.); antonio.polimeni@roma1.infn.it (A.P.); 4School of Chemistry, University of Nottingham, Nottingham NG7 2RD, UK; joseph.glover@nottingham.ac.uk (J.G.); elena.besley@nottingham.ac.uk (E.B.); 5Institute for Problems of Materials Science, National Academy of Sciences of Ukraine, Chernivtsi Branch, 58001 Chernivtsi, Ukraine; zakhar.kovalyuk@gmail.com

**Keywords:** indium selenide, intercalation, Kaufman ion source, Raman, photoluminescence, van der Waals crystals, hydrogen

## Abstract

The emergence of the hydrogen economy requires development in the storage, generation and sensing of hydrogen. The indium selenide (γ-InSe) van der Waals (vdW) crystal shows promise for technologies in all three of these areas. For these applications to be realised, the fundamental interactions of InSe with hydrogen must be understood. Here, we present a comprehensive experimental and theoretical study on the interaction of γ-InSe with hydrogen. It is shown that hydrogenation of γ-InSe by a Kaufman ion source results in a marked quenching of the room temperature photoluminescence signal and a modification of the vibrational modes of γ-InSe, which are modelled by density functional theory simulations. Our experimental and theoretical studies indicate that hydrogen is incorporated into the crystal preferentially in its atomic form. This behaviour is qualitatively different from that observed in other vdW crystals, such as transition metal dichalcogenides, where molecular hydrogen is intercalated in the vdW gaps of the crystal, leading to the formation of “bubbles” for hydrogen storage.

## 1. Introduction

In the effort to tackle climate change dramatic decarbonisation must be pursued to decarbonise sectors, such as: energy, transport, heating, and the chemical industry. One proposal for meeting these goals is through the establishment of the hydrogen economy, a system wherein hydrogen displaces fossil fuels as an energy vector and chemical feedstock. However, while global demand for hydrogen continues to grow and more policy incentives for hydrogen technologies emerge [1], there remains technological barriers to its widespread adoption. These barriers include the development of performant hydrogen storage systems, efficient hydrogen generation, low-cost fuel cells, hydrogen sensing, hydrogen purification, and the associated safety platforms for implementation [2,3,4]. In these fields, van der Waals (vdW) layered crystals and their two-dimensional (2D) counterparts show promise. To facilitate the selection of such materials, fundamental research into the interaction between these materials and hydrogen is needed. This is done here for the vdw material γ-InSe.

The diverse family of 2D materials has grown rapidly since the successful exfoliation of graphene in 2004 [5]. Their reduced dimensionality restricts the motion of charge carriers to a plane. The resulting quantum confinement radically modifies optical and electronic properties, giving access to a wide range of new quantum phenomena of fundamental and applied interest, including low-power electronics, broad-band photo-detection and nanophotonics [6,7,8,9]. These properties exist alongside the good flexibility and strength inherent to 2D materials [10,11,12], with tuneability possible through strain engineering, alloying, and surface modification [13,14,15,16,17].

Van der Waals crystals are composed of stacked 2D monolayers, which are separated by inter-layer gaps, the so-called vdW gap, and are held together by vdW forces. The vdW gaps introduce a high degree of anisotropy to the crystal, alongside a significant unoccupied internal volume, and a large internal surface area. The presence of the vdW gap allows for the intercalation of guest species as a means to modify materials’ structural, electronic, and optical properties [18]. These properties, the large surface area to volume ratios, and the presence of the vdW gap [19,20,21,22,23,24] make vdW materials candidates for hydrogen technologies.

Of particular interest is the group III–VI layered semiconductor γ-InSe, which has gained attention due to its unique electronic band structure and interesting electrical and optical properties [25,26,27,28]. γ-InSe crystallises in the R3m space-group, with unit cell dimensions of a=b= 4.002 Å and c= 24.946 Å [29]. The crystal structure of γ-InSe can be seen in Figure 1a, where the vdW gap comprises approximately 40% of the structure by volume [30]. The optical, electronic and structural properties of InSe make its bulk and low dimensional forms interesting candidates for hydrogen storage, water splitting and gas sensing applications [30,31,32,33,34,35,36,37,38]. However, experimental research is still in its infancy. Previously, experiments have introduced hydrogen into γ-InSe electrochemically, reaching compositions of up to H_5_InSe [30,31,32,33,34]. It was suggested that molecular hydrogen was incorporated as a monolayer within the vdW gap [33]. Since the electrochemical incorporation of hydrogen in γ-InSe, the exposure of layered crystals to a low energy proton beam, generated using a Kaufman ion source, has yielded a surprising result. Upon exposure to the beam, bulk transition metal dichalcogenide (TMD) vdW crystals of WS_2_, WSe_2_, WTe_2_, MoS_2_, MoSe_2_ and MoTe_2_ formed domed structures on their surfaces [39,40]. These domes are microns in size and contain molecular hydrogen at 10s of bar, trapped beneath the uppermost monolayer of the crystal. Here, we examine the exposure of γ-InSe to hydrogen by a Kaufman ion source.

For a sample of γ-InSe exposed to hydrogen, we propose five different sites for hydrogen incorporation (Figure 1). In the first (Figure 1a), H_2_ molecules are formed but their presence is sufficiently unfavourable that they are expelled from the crystal, leaving behind unmodified γ-InSe. In the second, shown in Figure 1b, H_2_ molecules are formed and their presence is sufficiently favourable to remain in the crystal, forming a monolayer in the vdW gap. This could lead to the formation of bent layers or domes, as seen in other layered crystals [39,40]. In the third, shown in Figure 1c, H_2_ is not formed and instead atomic hydrogen bonds to the Se atoms within γ-InSe. In the fourth, shown in Figure 1d, Se vacancies present in γ-InSe are occupied by atomic hydrogen. In the fifth, Se vacancies present in γ-InSe are occupied by molecular H_2_.

We examine these different configurations by experiment and theory. Through a combination of photoluminesence (PL), Raman spectroscopy, X-ray photoelectron spectroscopy (XPS), density functional theory (DFT) simulations (including phonon mode calculations), and grand canonical Monte Carlo (GCMC) simulations, we propose that the formation of molecular hydrogen in the vdW gap of γ-InSe is unfavourable near ambient conditions. Instead, atomic hydrogen tends to bond preferentially to the InSe lattice modifying optical and vibrational properties of the crystal. In particular, hydrogen acts as a centre for non-radiative recombination of photogenerated carriers thus decreasing the PL intensity. In contrast, computational results indicate that high densities of molecular hydrogen would require a large increase in the size of the vdW gap, which could be potentially realised at high pressure and/or high temperatures. These results are pivotal to understanding the interaction between γ-InSe and hydrogen, consequently they are of importance when considering the use of γ-InSe in future experimental systems and eventual applications to the hydrogen economy.

## 2. Results

Samples of bulk γ-InSe were exposed to atomic hydrogen using a Kaufman ion source, an illustration of which can be found in Figure 2a. All results were obtained from samples exposed to low beam energies of either 10 eV or 32 eV at temperatures of $T_{H}=$ 25 °C and 80 °C, respectively. These samples were compared with pristine γ-InSe or samples covered by a H-opaque mask during the hydrogenation. A full description of hydrogenation conditions can be found in the methodology.

### 2.1. DFT

The incorporation of hydrogen in γ-InSe was examined by DFT (see methodology for details). The two main structures investigated by DFT are shown in Figure 1b,c. Upon introducing molecular hydrogen to γ-InSe, the size of the vdW gap increases significantly from 3.6 Å to 6.1 Å. The position of molecular hydrogen relative to the neighbouring γ-InSe layers are shown in the inset of Figure 1b. In contrast, atomic hydrogen binds to selenium introducing a distortion to the γ-InSe lattice and reducing the vdW gap from 3.5 Å to 2.6 Å. The addition of atomic hydrogen also increases the adjacent In-Se bond length from 2.68 Å to 2.83 Å. In both cases, the addition of hydrogen does not alter the stacking of the individual layers of γ-InSe.

The binding energies of H_2_ in different configurations of bulk and monolayer InSe, relative to separate InSe and H_2_ gas, are shown in Table 1. In bulk γ-InSe, both interlayer and intralayer incorporation of hydrogen is found to be energetically unfavourable. In monolayer InSe, intralayer incorporation of hydrogen is found to be unfavourable but adsorption of hydrogen molecules on the surface is favourable. As only weak vdW interactions are expected between InSe and H_2_, it might be expected from the calculated binding energies that molecular H_2_ is unlikely to be adsorbed in bulk γ-InSe.

### 2.2. Photoluminescence

In contrast to hydrogenated TMDs [39,40], no dome structures are formed on the surface of hydrogenated InSe (32 eV), as can be seen in an optical microscopy image shown in the inset in Figure 2b. Across multiple surface locations and multiple samples, no significant difference could be seen in the optical images between those samples exposed to the Kaufman ion source and the control samples of pristine and masked InSe. These optical investigations were conducted both immediately after exposure and after 12 days. The length scales investigated (down to a few hundred nanometers) would reveal the presence of dome structures of the size or larger than those reported in TMDs [39,40].

The room temperature PL spectra of masked and hydrogenated (32 eV) samples are shown in Figure 2b. The PL peak position remains unchanged between the two samples, at 1.25 eV, as do the PL emission lineshapes. Immediately after hydrogenation, the PL intensity of hydrogenated InSe (32 eV) was approximately two orders of magnitude lower than that of the masked sample. The reduction in PL intensity in hydrogenated InSe (10 eV) immediately after exposure was of approximately one order of magnitude. After 12 days the PL amplitude in hydrogenated InSe (32 eV) was approximately one order of magnitude less than in masked InSe. In hydrogenated InSe (10 eV) the PL amplitude remained approximately constant relative to the masked sample. Beyond 12 days the quenching of PL signals in both InSe samples hydrogenated at 10 eV and 32 eV remained approximately constant at one order of magnitude for a period of at least 5 months. The masked sample recorded no quenching in PL intensity as compared immediately before and after the exposure.

Low-temperature (T= 10 K) PL spectra are shown in Figure 2c,d for pristine and hydrogenated (10 eV) InSe, respectively. These spectra were gathered 5 months after the initial exposure to the Kaufman source. The multiple emission lines at low photon energies (<1.32 eV) are most prominent at low excitation powers and are associated with impurities and/or defects [41,42]. No features at photon energies less than 1.32 eV can be successfully attributed to hydrogen, as the variation between the two samples is indistinguishable from the variation found within the individual samples. For photon energies above 1.32 eV, the exciton (X), biexciton (XX) and exciton-exciton scattering (X-X) PL lines can be identified [41,42]. They dominate the PL spectra at high excitation powers. The X-peak in pristine and hydrogenated (10 eV) InSe is at 1.340 ± 0.002 eV and 1.337 ± 0.002 eV, respectively. The position of the XX-peak in pristine and hydrogenated (10 eV) InSe is at 1.333 ± 0.002 eV and 1.334 ± 0.002 eV, respectively. The position of the X-X peak in pristine and hydrogenated (10 eV) InSe is at 1.331 ± 0.002 eV and 1.332 ± 0.002 eV, respectively. The positions of the X and XX peaks in pristine γ-InSe are in good agreement with results from the literature [41,42]. However, the X-X peak position is at a higher energy in our sample than the 1.32 eV peak reported previously [41].

The insets in Figure 2c,d show the dependence on power (*P*) of the integrated intensity of the PL spectrum (*I*) in pristine and hydrogenated γ-InSe. This is described by a power law $I∼P^{\alpha }$ [43]. In both samples two distinct regions can be clearly seen. In the pristine sample, α increases from ∼1.0 to ∼2.8 with increasing power. A similar behaviour is observed in the hydrogenated sample with the value of α increasing from ∼1.0 to ∼2.3 going from low to high powers. The superlinear dependence at high power reflects the contribution to the PL from XX and X-X recombination. The onset of the superlinear dependence occurs at higher power in the hydrogenated sample than in the pristine sample. This suggests an increasing concentration of defects and/or impurities in these samples, causing the recombination of carriers from these states to acquire importance relative to that of free excitons. This finding and the lower room temperature PL intensity in the hydrogenated samples suggests that hydrogen acts as a centre for non-radiative recombination of carriers.

With increasing the sample temperatures from 10 K up to 300 K the energy peak position of the PL emission was found to vary continuously to its room temperature value of 1.25 eV (see Appendix A).

### 2.3. Raman

Raman spectroscopy was conducted on hydrogenated and masked samples 12 days after the exposure to the Kaufman ion source. Figure 3a shows the Raman spectra for masked and hydrogenated (32 eV) InSe. Three primary peaks can be seen at 118 cm^−1^, 180 cm^−1^, and 230 cm^−1^, corresponding to the $A_{1g}^{1}$, $E_{2g}^{1}$ and $A_{1g}^{2}$ vibrational modes of γ-InSe, respectively [44]. The positions of the three primary peaks are the same in the hydrogenated and masked samples. The tail of the first Raman mode is broadened in hydrogenated InSe towards lower Raman shifts. This broadening is consistent throughout different positions of the sample. Both samples gave a similar intensity, with the intensity of hydrogenated InSe (32 eV) reduced by a factor of 1.4. Raman spectra were also gathered over an extended range, up to 4200 cm^−1^, but no additional features were observed in any sample. Raman spectra were measured over a period of 5 months showing reproducible features.

To assess whether the measured changes in the Raman spectra are associated with the incorporation of hydrogen into the crystals, we calculated the phonon modes by DFT (see methodology for details). Figure 3b,c show the simulated phonon density of states for a pristine InSe monolayer and an InSe monolayer with atomic hydrogen bonded to Se. The addition of atomic hydrogen broadens the peaks in the 140–190 cm^−1^ range. The total number of modes remains unchanged over the shown range. The position of the first Raman active mode shifts from 102 cm^−1^ to 99 cm^−1^ upon the addition of atomic hydrogen. The second phonon mode is at 164 cm^−1^ in the pristine monolayer but could not be successfully identified in the hydrogenated monolayer. The third phonon mode shifts from 221 cm^−1^ to 215 cm^−1^ upon the addition of hydrogen.

Phonon calculations were also conducted by DFT on pristine bulk γ-InSe and bulk γ-InSe with molecular hydrogen included in the vdW gap. The corresponding phonon density of states is shown in Figure 4a,b. The peaks in the phonon density of states are noticeably narrowed between 150 cm^−1^ and 190 cm^−1^ in the hydrogenated structure. A new peak emerges between 230 cm^−1^ to 233 cm^−1^ associated with translational modes of hydrogen within the vdW gap. The vibrations corresponding to the three primary Raman modes in InSe are marked in Figure 4a,b. The position of the first primary phonon mode increases from 98 cm^−1^ in the pristine sample to 100 cm^−1^ in the hydrogenated sample. The position of the second primary phonon mode decreases from 163 cm^−1^ in the pristine sample to 161 cm^−1^ in the hydrogenated sample. The position of the third primary phonon mode shifts from 206 cm^−1^ in the pristine sample to 214 cm^−1^ in the hydrogenated sample.

The experimental broadening of the first Raman peak is best described by the atomic hydrogen structure. This structure successfully predicts a shift of the $A_{1g}^{1}$ mode towards a lower shift. In contrast, the structure containing molecular hydrogen suggests a shift of the mode in the direction opposite to that observed experimentally. It should be noted that the change recorded in the phonon positions is approaching the limit of our current computational setup. The true value of the change may be different from those generated by DFT. However, the result should provide the correct qualitative prediction.

### 2.4. Grand Canonical Monte Carlo Simulations

Finally, to assess whether the incorporation of molecular hydrogen is favourable, we examined the computed “heatmaps” of hydrogen density in γ-InSe with differently sized vdW gaps at a pressure of 20 bar at T= 298 K. The “heatmaps” in Figure 5 indicate the favourability for hydrogen molecules to reside at different locations within and outside the crystal. Without changing the size of the vdW gap of γ-InSe, no molecular hydrogen is intercalated (Figure 5a). Upon increasing the vdW gap by 2 Å, some hydrogen is incorporated within the gap, with the majority remaining outside the crystal (Figure 5b). Upon increasing the gap by 4 Å, there is a much higher density of hydrogen within the crystal, at a higher density than outside the crystal (Figure 5c). Results across a larger number of vdW gap spacings are available in the Appendix A. These studies indicate that significant hydrogen accumulation is achieved for gap increases of above 2.5 Å over pristine γ-InSe. This compares with the kinetic diameter of the H_2_ molecule of 2.9 Å [45].

### 2.5. XPS

An important consideration with InSe is its propensity to forming Se vacancies [46]. It was therefore necessary to consider a scenario whereby exposure to the proton beam would create Se vacancies. These vacancies would then be rapidly oxidised when exposed to air. XPS can be used to identify any additional oxygen present on the hydrogenated sample, as this would cause a shift towards higher binding energy [46]. Detailed scans of the In 3d and Se 3d peaks can be seen in Figure 6a,b for both pristine γ-InSe and hydrogenated γ-InSe (32 eV). The In 3d doublet peaks are at binding energies of 452.5 eV and 445 eV for both the pristine and hydrogenated samples. The Se 3d doublet peaks are identified at 55.1 eV and 54.3 eV for both the pristine and hydrogenated samples. The low binding energy tail of hydrogenated γ-InSe is broadened in both In 3d and Se 3d doublets. For both In 3d and Se 3d doublets no broadening or additional features were observed towards higher binding energy. This behaviour was replicated in three detailed scans of the samples. The observed broadening is inconsistent with the oxidation of Se vacancies [46].

## 3. Discussion

Two possibilities must be entertained before discussing the significance of the results presented. The first is whether it is the exposure to the proton beam rather than the high sample temperatures ($T_{H}$ > 25 °C) that resulted in the observed changes. The second is whether hydrogen incorporation resulted from exposure of the crystal to the beam. On the first point, the PL and Raman results comparing exposed and masked samples can be considered. In this case the only difference between samples is exposure to the proton beam and it can be reasonably concluded that the observed quenching of the PL signal and broadening of the lowest measured Raman mode is due to the beam. On the second point, the XPS should be considered. The broadening of In 3d and Se 3d modes towards lower binding energy is inconsistent with oxidation of Se vacancies [46], the candidate mechanism through which the beam could damage γ-InSe. It is then reasonable to assume that hydrogen incorporation is the origin of the observations.

The next consideration is whether the hydrogen could be included in γ-InSe in its molecular form. The Raman results provide the best probe for this possibility. Gaseous hydrogen is Raman active in the 4100 cm^−1^ to 4200 cm^−1^ range [47]. In a crystal structure this range can shift [48,49,50]. If molecular hydrogen is present in the samples then a Raman peak would be expected in the 3800 cm^−1^ to 4200 cm^−1^ range. None were observed.

The broadening of the first Raman mode, observed in Figure 3a, is not replicated in the simulated phonon density of states of H_2_ intercalated γ-InSe shown in Figure 4b. The position of the $A_{1g}^{1}$ mode instead shifts to a higher value rather than the experimentally observed lower value. Upon the addition of molecular hydrogen, the phonon density of states instead becomes more like that of an InSe monolayer, as the $A_{1g}^{1}$ and $A_{1g}^{2}$ modes move towards their monolayer values and there is a narrowing of the peaks in the 150 cm^−1^ and 190 cm^−1^ range. This can be understood due to the increased size of the vdW gap, reducing the strength of the interlayer coupling. This is untrue for the $E_{2g}^{1}$ mode, which shifts away from its value in the monolayer.

The GCMC simulations shown in Figure 5 illustrate the extent to which molecular hydrogen incorporation within γ-InSe is unfavourable. The vdW gap requires an increase in size of at least 2.5 Å for significant molecular hydrogen incorporation to occur. This is in agreement with the DFT calculations of the molecular hydrogen binding energy in γ-InSe, which shows that molecular hydrogen incorporation within the vdW gap is unfavourable. These results indicate that the hydrogen is not included in γ-InSe as H_2_. Our experiments with a Kaufman ion source are further supported by preliminary studies on the direct incorporation in γ-InSe of molecular hydrogen at pressures of up to 60 bar at room temperature. Also, in this case no significant incorporation of molecular hydrogen was observed.

Dismissing molecular hydrogen as the state of incorporation also precludes the possibility of hydrogen filled domes, as seen in TMD crystals [39,40]. This is further supported by the lack of any domes visible by optical microscopy. The domes would also be expected to modify the PL spectra, as quantum confinement and/or strain effects would be observed in the uppermost monolayer of the crystal. In the hydrogenated samples the PL peak shape and position remained unchanged.

The only proposed state capable of qualitatively reproducing the broadening of the first Raman mode is that of atomic hydrogen bonded to the γ-InSe structure. The $A_{1g}^{1}$ mode moves in the correct direction upon hydrogenation, as shown in Figure 3b,c. This shift can be understood due to the additional mass of the bonded hydrogen atom and a softening of In-Se bonds adjacent to the H-Se bond. The $A_{1g}^{2}$ mode is shifted in the theoretical results but not the experimental data. This may be due to limits in computational methods, which necessitate high hydrogen densities. The state corresponding to atomic hydrogen binding to γ-InSe state is considered the most likely state of hydrogen in the hydrogenated samples.

The proposed site of hydrogen incorporation contrasts with that of molecular hydrogen in electrochemical studies [30,31,32,33,34]. These differing results form part of a larger H-γ-InSe phase space, alongside the other proposed states in this paper. An understanding of this behaviour is crucial for the implementation of γ-InSe technologies for use with hydrogen, as vibrational spectra and thermodynamic simulations have shown, these states have qualitatively different behaviour. Why molecular hydrogen was not formed in γ-InSe exposed to the Kaufman ion source remains an open question. However, potential candidates include: the strength of the H-Se bond, a low flux of atomic hydrogen, poor diffusion of atomic hydrogen, and the large vdW gap required for significant molecular hydrogen incorporation.

In the proposed state, the bound hydrogen can be said to act as a centre for non-radiative recombination of charge carriers. This conclusion is supported by the quenching of the room temperature PL and the weaker power dependence of the excitonic lines in hydrogenated γ-InSe than pristine γ-InSe. The small modification in the energy position of the exciton peak found between hydrogenated and pristine InSe is thought to be unreliable. This is due to the difficulty in determining the position of the X peak and the unchanged positions of the XX and X-X features.

Whilst it is suggested that hydrogen introduced by a Kaufman ion source to γ-InSe remains in its atomic form, it is recognised that the conditions investigated remain limited and the proposed five states may be achievable through other means. The qualitative difference shown in the phonon density of states of molecular and atomic hydrogen provide a means to probe the state of hydrogen incorporation. The introduction of molecular hydrogen is suspected to require a large increase in the vdW gap, for example at high pressures and/or temperatures.

## 4. Materials and Methods

The γ-InSe crystal was grown using the Bridgman method from a non-stoichiometric melt of In_1.03_Se_0.97_. All hydrogenated and control samples were cleaved from the same bulk ingot. The sample surfaces were refreshed via exfoliation immediately before exposure to the hydrogen beam.

### 4.1. Hydrogenation

The samples were exposed to a proton beam of uniform energy by a Kaufman ion source, a representation of which is shown in Figure 2a. Hydrogen gas is ionised in an ionisation chamber before being accelerated towards the samples by a series of grids. The samples were mounted using silver paste to the sample plate, which is grounded to prevent charging. A heating lamp is positioned behind the sample plate. A probe positioned in the path of the ion beam determines the ion flux. The sample chamber is pumped down to a vacuum better than 10^−6^ mbar before exposure to H^+^. The results from two hydrogenated samples are presented in this paper. The first was exposed to a beam energy of 32 eV at a temperature of $T_{H}$ = 80 °C, with a total attained dose of 6 × 10^16^ ions cm^−2^. The second sample was exposed to a beam energy of 10 eV at a temperature of $T_{H}$ = 25 °C, with a total attained dose of 10^15^ ions cm^−2^. A masked sample was positioned alongside the 32 eV sample in the chamber but was shielded from the beam.

### 4.2. Photoluminescence and Raman

The PL and Raman spectra were measured using a frequency doubled Nd:YVO_4_ laser with a wavelength of 532 nm focused onto the sample using a confocal microscope through a 100× objective. The laser spot was approximately 1 μm in diameter. The PL measurements used a 150 lines/mm grating and the Raman measurements used a 1200 lines/mm grating. Detection was performed by a linear CCD array. Low temperature measurements were conducted using a cold finger cryostat under constant gas flow conditions, enabling temperatures between 10 K and 300 K to be achieved with a variation of less than 1 K at all temperatures. The laser spot was approximately 5 μm in diameter in the low temperature measurements.

### 4.3. XPS

The XPS studies on the samples were performed using a Kratos AXIS ULTRA with a mono-chromated Al kα X-ray source (1486.6 eV) operated at 10 mA emission current and 12 kV anode potential (120 W). Sample charging was minimised using a charge neutraliser filament. A hybrid aperture was used with a spot area of approximately 300 × 700 μm. The detailed scans on the peaks of interest were conducted using a 20 eV pass energy and a resolution of 100 meV. An acquisition time of 10 minutes was used. Three detailed scans were conducted on the In 3d and Se 3d peaks of all samples.

### 4.4. DFT Calculations

All density functional theory calculations have been performed using the CP2K code, which uses a mixed Gaussian/plane-wave basis set [51,52]. We employed triple-ζ polarization quality Gaussian basis sets [53] and a 600 Ry plane-wave cutoff for the auxiliary grid, in conjunction with the Goedecker-Teter-Hutter pseudopotentials [54,55]. A convergence threshold of 1.0 × 10^−7^ Hartree was used for the self-consistent field (SCF) cycle; structural optimizations were considered as converged when the maximum force on atoms fell below 4.5 × 10^−4^ Hartree/Bohr. All calculations were performed in the Γ-point approximation for sufficiently large cells. All DFT calculations, including single point energies, geometry/cell optimizations and phonon calculations, were performed using the PBE functional [56], with Grimme’s D3 van der Waals correction (PBE+D3) [57]. The phonon calculations were performed using the finite-displacement method with tighter SCF (1.0 × 10^−8^ Hartree) and force (1.0 × 10^−4^ Hartree/Bohr) convergence criteria. In the finite-displacement method, the force constant, i.e the second derivatives of the total energy with respect to the atomic displacements, were computed numerically, and then the normal mode frequencies were obtained by diagonalization of the force constants matrix. In the current study, an atomic displacement of 0.01 Bohr was chosen to construct the force constants matrix.

The assignment of the phonon modes in the phonon density of states plots was done manually. For pristine γ-InSe, the calculated phonon modes were first compared to the known Raman active modes of γ-InSe [44,58]. Specific modes were then identified visually in hydrogenated γ-InSe (see animations in the Appendix A).

### 4.5. Grand Canonical Monte Carlo Simulations

To simulate single-component adsorption isotherms, GCMC simulations were employed using the RASPA software package at 298 K [59]. Input fugacities were calculated using the Peng-Robinson equation of state [60]. For each point on the isotherm, the system was allowed to equilibrate using 10,000 cycles. Averages such as the number of molecules adsorbed were then recorded over a further 10,000 cycles. In this work, 1 cycle = max(N, 50) random perturbations, where N is number of adsorbates in the simulation box. Perturbations included random translation, rotation, insertion and deletion of adsorbates. All adsorbents were modelled using the general Universal Force Field as rigid species (atoms fixed in their crystallographic positions) [61]. In each simulation, a vacuum of at least 15 Å was added above and below the adsorbent along the c-axis. Hydrogen adsorbates were described using a rigid 3-site model consisting of two hydrogen sites bonded via a centre of mass site [62]. The adsorbent and adsorbate force-fields are both comprised of a 12-6 Lennard-Jones potential which employ the Lorentz-Berthelot mixing rules to determine any cross-term interactions [63,64,65]. All Lennard-Jones interactions were calculated using a cut-off of 12.8 Å and electrostatic interactions between adsorbates were handled using the Ewald summation technique [66]. A 7 × 7 × 1 supercell was employed to comply with the minimum image convention. Adsorbent-adsorbate electrostatic interactions were assumed negligible and therefore removed from the simulation to increase computational efficiency. All force-field parameters used in this work are summarised in Table 2.

The choice of the UFF parameters was based on its successful application to studying the adsorption properties of porous solids such as metal organic frameworks [67,68]. We believe that this choice allows for qualitative insights, including the general adsorption sites and the ability for H_2_ to be stored within the layers of these materials.

## 5. Conclusions

Samples of γ-InSe were irradiated with protons from a Kaufman ion source. The suppressed room temperature PL intensity and decreased excitonic character at low temperature indicate that hydrogen acts as a centre for non-radiative recombination of photo-generated carriers. The incorporation of hydrogen modifies the 118 cm^−1^ Raman mode of γ-InSe. This modification can be explained through the binding of atomic hydrogen to the selenium atoms in the crystal. DFT-calculated phonon density of states indicated the ability for molecular hydrogen to decouple the layers of γ-InSe. This is explained by the large increase in the vdW gap of at least 2.5 Å required for significant molecular hydrogen incorporation. The shifting of the 118 cm^−1^ Raman mode in opposite directions for the atomic hydrogen and molecular hydrogen structures allows for distinguishing the two states. This combined with the modifications introduced by both, constitute important results for the integration of γ-InSe into hydrogen technological applications.

## Figures and Tables

**Figure 1 molecules-25-02526-f001:**
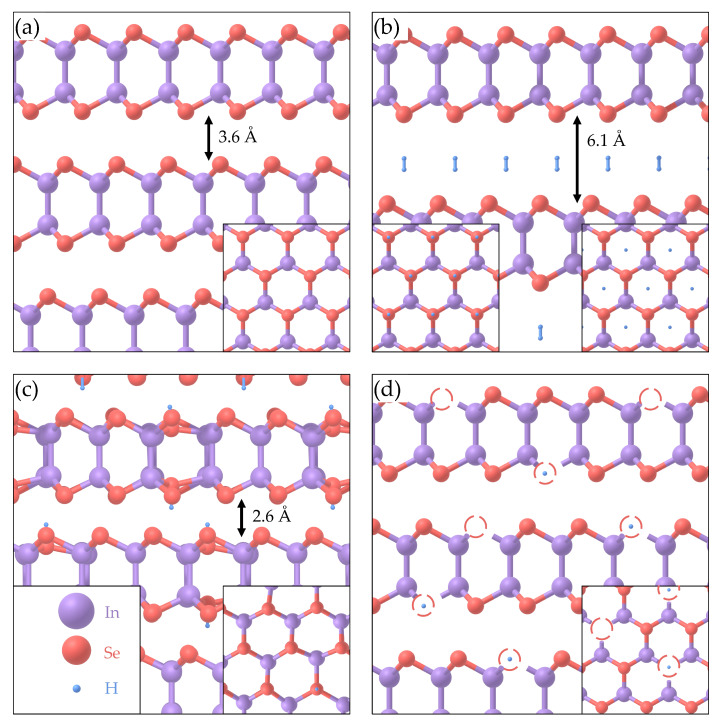
Crystallographic structures for: (**a**) Pristine bulk γ-InSe viewed in plane and (inset) out of plane. (**b**) Bulk γ-InSe containing H_2_, as computed using density functional theory (DFT), viewed in plane and (inset) out of plane. (**c**) Bulk γ-InSe with atomic hydrogen bonded to Se, as computed using DFT, viewed in plane (inset) out of plane. (**d**) Proposed structure of bulk γ-InSe with Se vacancies, some of which contain atomic hydrogen, as viewed in plane and (inset) out of plane.

**Figure 2 molecules-25-02526-f002:**
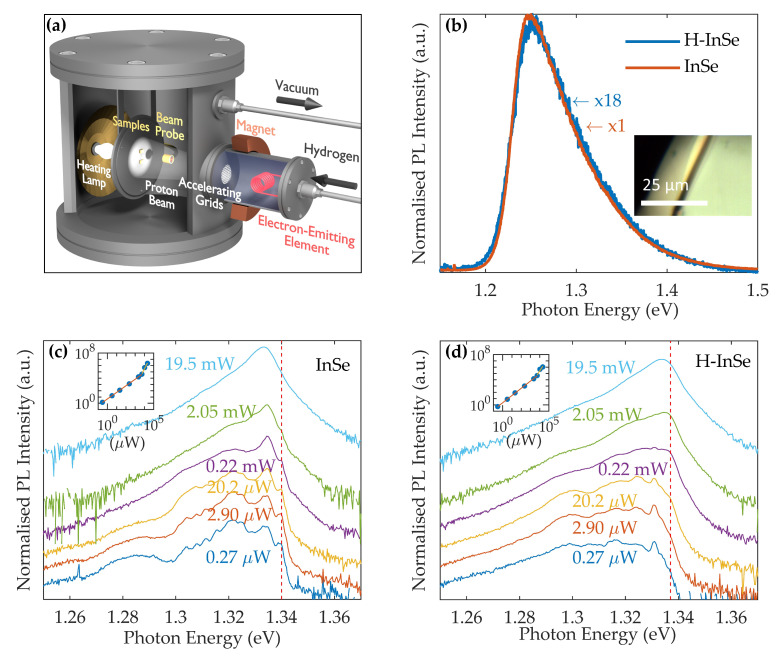
(**a**) Schematic of the Kaufman ion source for incorporation of hydrogen into γ-InSe. (**b**) Normalised photoluminesence (PL) spectra measured in masked and hydrogenated (32 eV) γ-InSe, 12 days after hydrogenation (λ= 532 nm, P= 0.2 mW, T= 300 K). Inset: optical microscopy image of the surface of hydrogenated (32 eV) γ-InSe. (**c**,**d**) PL spectra measured at different exciting laser powers in pristine and hydrogenated (10 eV) γ-InSe (λ= 532 nm, T= 10 K). The spectra are plotted on a logarithmic scale with the dashed lines denoting the energy position of the exciton line (X). Insets: dependence of the total integrated PL signal on excitation power (P) on a logarithmic scale.

**Figure 3 molecules-25-02526-f003:**
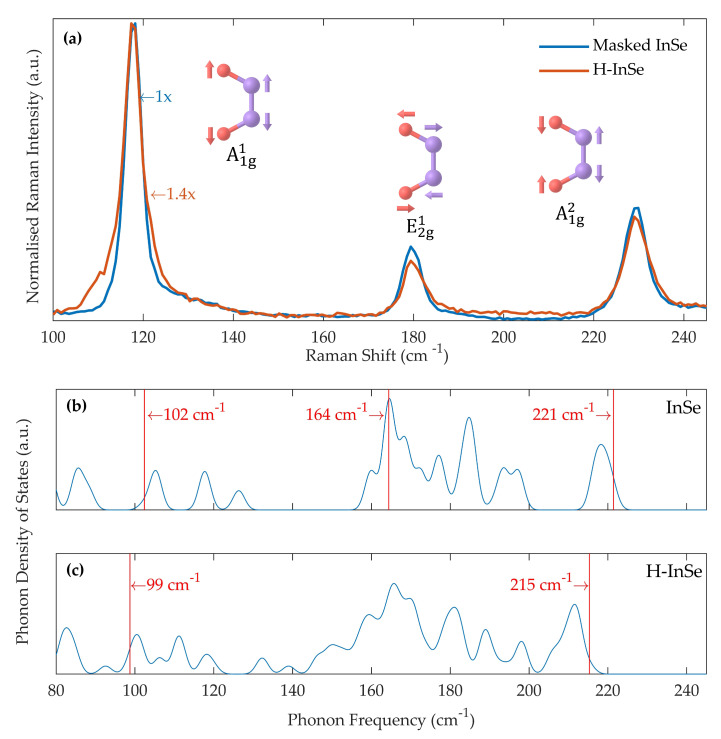
(**a**) Normalised Raman spectra of masked and hydrogenated (32 eV) γ-InSe, measured 12 days after hydrogenation (λ= 532 nm, P= 0.2 mW, T= 300 K). The spectra are normalised to the first Raman peak. Insets: vibrational modes responsible for the three peaks. (**b**,**c**) DFT-calculated phonon density of states for: (**b**) a pristine InSe monolayer and (**c**) atomic hydrogen bonded to an InSe monolayer. These were calculated using a 4 × 4 × 1 supercell. The phonon density of states are Gaussian broadened with σ= 1.5 cm^−1^. Vertical red lines in (**b**,**c**) mark the position of Raman active modes, these correspond to the peaks in (**a**).

**Figure 4 molecules-25-02526-f004:**
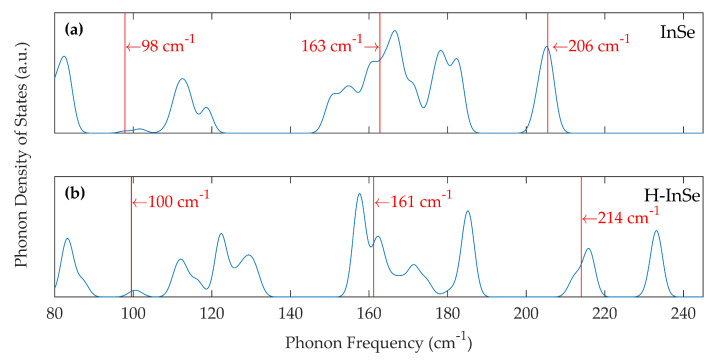
DFT-calculated phonon density of states in: (**a**) Bulk pristine γ-InSe, using a 3 × 3 × 2 supercell. (**b**) Bulk γ-InSe intercalated with H_2_, as shown in Figure 1b, using a 3 × 3 × 2 supercell. The phonon density of states are Gaussian broadened with σ= 1.5 cm^−1^. Vertical red lines in (**a**,**b**) mark the position of Raman active modes. These correspond to the peaks in Figure 3a.

**Figure 5 molecules-25-02526-f005:**
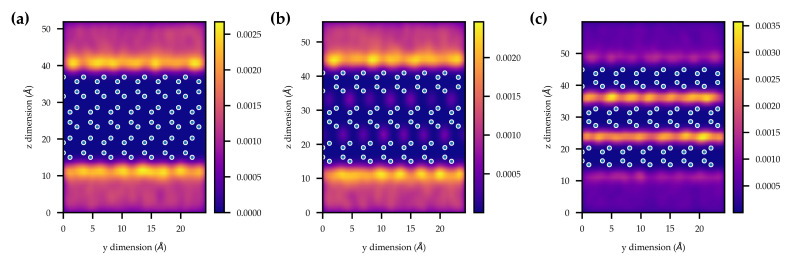
H_2_ density “heatmaps” created using kernel density estimation from grand canonical Monte Carlo (GCMC) simulations at 20 bar. (**a**) Pristine γ-InSe. (**b**) γ-InSe with the vdW gaps increased by 2 Å. (**c**) γ-InSe with the vdW gaps increased by 4 Å. The open circles represent the In and Se atomic positions.

**Figure 6 molecules-25-02526-f006:**
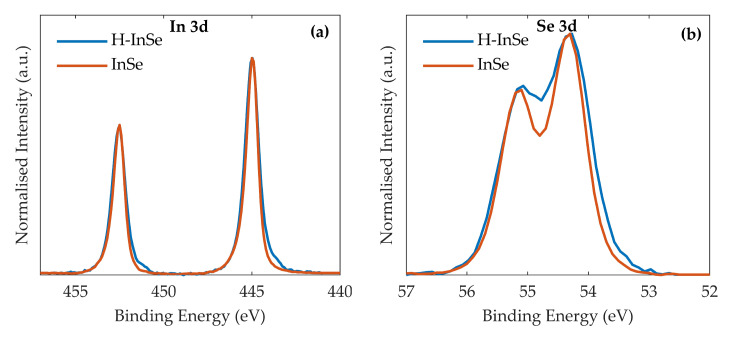
Charge corrected and normalised X-ray photoelectron spectroscopy (XPS) spectra for pristine and hydrogenated (32 eV) γ-InSe. Shown are the regions corresponding to: (**a**) The In 3d doublet. (**b**) The Se 3d doublet.

**Table 1 molecules-25-02526-t001:** Binding energies of H_2_ and InSe in different configurations, as determined using DFT. Binding energies are measured in kJ mol^−1^. For all four cases, we consider the adsorption of a single H_2_ molecule in a sufficiently large supercell. Only the atomic positions were allowed to relax while the cell parameters were kept fixed. For interlayer incorporation of a single H_2_ molecule in bulk InSe, we performed an additional calculation in which we also relaxed the cell parameters. The calculated binding energy is shown in brackets.

γ **-InSe Bulk**	**Interlayer**	**Intralayer**
	30.1 (29.3)	19.8
**InSe Monolayer**	**On Surface**	**Intralayer**
	−6.0	20.7

**Table 2 molecules-25-02526-t002:** A list of force-field parameters used in this work.

Atom	σ(Å)	ϵ(K)	q(e)
In	4.463	301.43	-
Se	4.205	146.44	-
H_H_2_	0.000	0.000	0.4829
COM_H_2_	2.958	36.700	−0.9658

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
