# Peer review of "The Interaction of Hydrogen with the van der Waals Crystal γ-InSe"

_molecules, 2020, doi:10.3390/molecules25112526_

Round 1
Reviewer 1 Report
The manuscript investigates the interaction of InSe with hydrogen using a combined experimental and theoretical approach. The InSe samples have been irradiated with a proton beam produced in a Kaufman ion source. This approach has shown very interesting results in recent experiments with Transition Metal Dichalcogenides. In these materials, proton irradiation induces the creation of nano/micro-domes filled up with Hydrogen. This effect strongly modifies the photoelectronic properties of the compounds. In the present case with InSe, however, this effect is not observed. Proton irradiation has a “weak” effect on the properties of InSe, leading to the quenching of the room temperature photoluminescence signal and a modification of the vibrational modes of InSe. Experiments are complemented with a theoretical modelling by density functional theory and gran canonical Monte Carlo simulations. According to the experimental observations and the theoretical modelling, it is concluded that proton irradiation of InSe produces defects related to the incorporation of atomic Hydrogen into the InSe crystal. The present work has been well planned and conducted and it deserves to be published in Molecules. There are only two minor questions that should be addressed before its publication:
-The introduction is strongly focused on applications related to hydrogen generation, storage and sensing. Although there is some interest in these fields, the relationship of the present results with these areas is not clear enough. The possible application of the present results in these fields is dubious. I suggest to modify the introduction, reducing the weight of the possible applications in these fields and focusing the article using a more fundamental perspective.
-Reference list must be checked. There are some mistakes (year in reference 5, article title in reference 21, …)
Reviewer 2 Report
Review of manuscript molecules-805175 ("The Interaction of Hydrogen with the
van der Waals Crystal γ-InSe" by James Felton et al.):
The manuscript provides results and discussion of the combined experimental and
computational (density functional theory (DFT)-based) study of interaction of
hydrogen and γ-InSe.
Subject of the present study is interesting and the methodology used is
appropriate. Discussion is clear and this manuscript is, essentially, suitable
for publication.
I would like to propose minor corrections/additions prior to the publication
in Molecules.
More detailed suggestions are given below:
#1
In lines 163-164 it was stated that the second phonon mode calculated by DFT
for hydrogenated monolayer could not be successfully identified.
It would be instructive if more details could be provided on
assignation of DFT calculated phonon modes to the experimental ones.
#2
Universal Force Field (UFF) was employed in the grand canonical Monte Carlo
simulations.
Additional references could be provided to justify employment of the UFF in the
present work.
Reviewer 3 Report
Overall, it is a good paper.
Just a small question regarding the peculiar shift of A11g peak in Raman spectra. Does the peak shift increase with the increase of hydrogen loading? Presenting the peak shift against the hydrogen loading might be helpful for further discussion. It is, however, only a suggestion.
